# Guarantees of confidentiality via Hammersley-Chapman-Robbins bounds

**Kamalika Chaudhuri**                                                    *kamalika@meta.com*
*Fundamental Artificial Intelligence Research*
*Meta Platforms, Inc.*

**Chuan Guo**                                                              *chuanguo@meta.com*
*Fundamental Artificial Intelligence Research*
*Meta Platforms, Inc.*

**Laurens van der Maaten**                                           *lvdmaaten@meta.com*
*Fundamental Artificial Intelligence Research*
*Meta Platforms, Inc.*

**Saeed Mahloujifar**                                                   *saeedm@meta.com*
*Fundamental Artificial Intelligence Research*
*Meta Platforms, Inc.*

**Mark Tygert**                                                           *mark@tygert.com*
*Fundamental Artificial Intelligence Research*
*Meta Platforms, Inc.*

**Reviewed on OpenReview:** *https://openreview.net/forum?id=DOWSP7y2cu*

## Abstract

Protecting privacy during inference with deep neural networks is possible by adding Gaussian noise to the activations in the last layers prior to the final classifiers or other task-specific layers. The activations in such layers are known as "features" (or, less commonly, as "embeddings" or "feature embeddings"). The added noise helps prevent reconstruction of the inputs from the noisy features. Lower bounding the variance of every possible unbiased estimator of the inputs quantifies the confidentiality arising from such added noise. Convenient, computationally tractable bounds are available from classic inequalities of Hammersley and of Chapman and Robbins — the HCR bounds. Numerical experiments indicate that the HCR bounds are on the precipice of being effectual for small neural nets with the data sets, "MNIST" and "CIFAR-10," which contain 10 classes each for image classification. The HCR bounds appear to be insufficient on their own to guarantee confidentiality of the inputs to inference with standard deep neural nets, "ResNet-18" and "Swin-T," pre-trained on the data set, "ImageNet-1000," which contains 1000 classes. Supplementing the addition of Gaussian noise to features with other methods for providing confidentiality may be warranted in the case of ImageNet. In all cases, the results reported here limit consideration to amounts of added noise that incur little degradation in the accuracy of classification from the noisy features. Thus, the added noise enhances confidentiality without much reduction in the accuracy on the task of image classification.

## 1 Introduction

The Hammersley-Chapman-Robbins (HCR) bounds of Hammersley (1950) and Chapman & Robbins (1951) provide easily interpretable, tight data-driven guarantees on confidentiality. The interpretation is especially simple and direct, lower bounding the variance of any estimator for reconstructing inputs to inference with

neural networks. Moreover, computing the bounds is efficient and straightforward. Confidentiality stems from suitable addition of noise; the HCR bounds quantify the effect of the added noise on the minimum possible variance of any estimator.

This paper studies the privacy preservation arising from adding independent and identically distributed Gaussian noise to the activations in the final layers of deep neural networks, that is, to the layers immediately preceding the classifiers used during supervised training of the nets and used for classification during inference. The most common terminology for these activations is "features" (or vectors of features). Less common synonyms are "feature embeddings" or simply "embeddings." Adding noise is also known as "dithering." Dithering features is a canonical method for limiting the quality of possible reconstructions of the inputs that generated the noiseless features.

The present paper omits consideration of adding noise directly to the data or for proxies to that process, such as DP-SGD of Abadi et al. (2016), since quantifying their privacy preservation in terms of the variance minimized over all possible estimators is trivial, given directly by the amount of noise added.

A method for quantifying privacy that is closely related to the HCR bounds is to use Fisher information and the Cramér-Rao bound, as advanced by Hannun et al. (2021) and others. The Cramér-Rao bound is most useful when a quadratic form specified by the Fisher information matrix is a good approximation to the expected loss (the risk function) near the parameters at which the Fisher information is evaluated. In contrast, the HCR bounds used below are typically tight whether or not a quadratic form specified by Fisher information evaluated at a single setting of parameters is a good approximation. Furthermore, evaluating the Fisher information for complicated models in machine learning such as deep neural networks can be difficult or costly at scale. The approach proposed below avoids most of the difficulties and is computationally tractable. In fact, the HCR bounds always exist, whereas the Cramér-Rao bounds exist only when the loss is sufficiently smooth.

The present paper could be viewed as providing an alternative to the Cramér-Rao bounds that is more practical. Indeed, the Cramér-Rao bound is a certain limit of an HCR bound. Subsection 2.5 below details the connection, a connection first made in the original works of Hammersley (1950) and of Chapman & Robbins (1951).

Unfortunately, the experimental results reported below indicate that the HCR bounds are weak for one of the data sets tested, at least with some standard neural nets for image classification (image classification is also known as "image recognition"). Section 4 below concludes that dithering features and leveraging the associated HCR bounds would be most useful in conjunction with cruder, brute-force methods for enhancing confidentiality of the inputs to inference (such as limiting the sizes of the vectors of features being revealed). From the point of view of actual practice, the idea proposed in the conclusion, Section 4, is to restrict the dimensions of the vectors of features accessible to an adversary, which directly limits the information available to the adversary. The HCR bounds are obviously more effective for practical applications in which the amount of information available to an adversary is limited. The numerical results reported in Section 3 below illustrate how the HCR bounds are more effective with smaller sizes of the vectors of features.

Earlier work, notably the thesis of Alisic (2021), also applies HCR bounds to quantify privacy, comparing favorably with the differential privacy of Dwork & Roth (2014). The focus of Alisic (2021) and the subset of the thesis reported by Alisic et al. (2020) is on confidentiality in the measurement of dynamical systems — quite different from the setting considered in the present paper — yet the results are complementary and consonant with those of the present paper.

The differential privacy of Dwork & Roth (2014) focuses on anonymization and preventing re-identification, rather than on the full confidentiality that directly prevents accurate reconstructions of input data (which is the focus of the present paper). Differential privacy provides guarantees in the form of parameters often denoted "$\varepsilon$" and "$\delta$," whereas the HCR bounds directly guarantee a minimum level of noise on the best-possible reconstructions obtainable from the dithered features. Appreciating the meanings of parameters such as $\varepsilon$ and $\delta$ from the usual formulations of differential privacy may require familiarity with mathematics that ordinary users are unlikely to have mastered, such as the exponential function or the concepts of relative sizes of probabilities or of the probabilities arising from classification via randomized algorithms. The HCR

bounds characterize via variance the minimum level of noise that an adversarial reconstruction could possibly attain; variance (or standard deviation) is a statistic measuring noise level, variability, spread, or dispersion that is meaningful to virtually anyone.

And for those who cannot understand the notion of "variance" or who prefer to see the quality of possible reconstructions displayed visually, Figures 2 and 4 below illustrate the best-possible adversarial reconstructions. Users are welcome to look at the pictures in Figures 2 and 4 — the noisy versions are the best any adversary who lacks prior information about the input can reconstruct the noiseless inputs. (Lacking prior information refers to the restriction that the adversary use an estimator which is statistically unbiased; of course, if the adversary knew all the pixels of the input image ahead of time via external sources of information, then a biased estimator that simply ignores all data and uses the prior information would completely compromise confidentiality. The general mathematical analysis developed in Section 2 below in principle allows for adjusting the bounds to account for prior information, though working out the full theory is well beyond the scope of the present paper.) Figures 2 and 4 effectively depict error bars on the best-possible reconstructions that any adversary could ever make.

Copious advantages of Cramér-Rao bounds over the canonical formulations of differential privacy have been detailed by Hannun et al. (2021) and others. Many of these advantages pertain to the HCR bounds, too. For instance, the HCR bounds adapt to the given data, whereas the standard formulation of differential privacy provides only worst-case guarantees, valid uniformly over all possible data sets (not only for the actual data of interest). The HCR bounds thus can be tighter and more powerful than the guarantees from typical differential privacy, at least with regard to preventing reconstruction of the data. Moreover, the HCR bounds can address individual pixels of images, individual modes in Fourier representations, or other parts of input data, whereas differential privacy typically operates at a coarser granularity ("membership inference").

The remainder of the present paper has the following structure: Section 2 develops theory and algorithms based on HCR bounds. Section 3 reports numerical experiments with the popular data sets MNIST, CIFAR-10, and ImageNet-1000 for image classification, when processed with standard architectures such as ResNets and Swin Transformers as well as with some especially simple, illustrative neural nets.[1] Section 4 draws several conclusions and suggests coupling the methods presented in the present paper with cruder, brute-force techniques for enhancing confidentiality.

## 2 Methods

This section details the methodology of the present paper. Subsection 2.1 briefly reviews the general formulation of Hammersley-Chapman-Robbins (HCR) bounds. Subsection 2.2 specifies the HCR bounds for dithering vectors of features specifically and develops algorithms for computing the bounds. Subsection 2.3 specializes the HCR bounds to the addition (to the features) of independent and identically distributed noise. Subsection 2.4 performs the calculations for the case that the distribution of the noise is a centered Gaussian. Subsection 2.5 considers a limit in which the HCR bounds become the classical Cramér-Rao bounds (under suitable assumptions of regularity in the parametric model, so that the relevant derivatives exist). The following section, Section 3, tests the bounds with several standard data sets and machine-learned models, when adding independent and identically distributed Gaussian noise to the features.

### 2.1 Hammersley-Chapman-Robbins bounds

This subsection reviews a classic bound introduced independently by Hammersley (1950) and Chapman & Robbins (1951). Specifically, we use the multivariate generalization detailed, for example, by Wikipedia contributors (2024).

We consider a family of probability density functions (pdfs) $f_\theta(x)$ of a vector $x$, parameterized by a vector $\theta$, with $x$ coming from an $n$-dimensional real vector space $\mathbb{R}^n$ and $\theta$ coming from a $p$-dimensional real vector

---

[1]Permissively licensed open-source codes that can automatically reproduce all results of the present paper are available at `https://github.com/facebookresearch/hcrbounds`

space $\mathbb{R}^p$. We consider any estimator $\hat{\theta}(X)$ of $\theta$; the estimator is a function of the vector $X$ of observations. We define $g(\theta)$ to be the expected value of $\hat{\theta}$ with respect to the pdf $f_\theta$, that is,

$$g(\theta) = \mathbb{E}_\theta[\hat{\theta}] = \int_{\mathbb{R}^n} \hat{\theta}(x)\, f_\theta(x)\, dx. \tag{1}$$

The Hammersley-Chapman-Robbins (HCR) bound is

$$\mathrm{Var}_\theta(\hat{\theta}_k) \geq \frac{(g_k(\theta + \varepsilon) - g_k(\theta))^2}{\mathbb{E}_\theta\left[\left(\frac{f_{\theta+\varepsilon}(X)}{f_\theta(X)} - 1\right)^2\right]} \tag{2}$$

for $k = 1, 2, \ldots, p$ and for any vector $\varepsilon$ in the same $p$-dimensional real vector space $\mathbb{R}^p$ to which $\theta$ belongs. In (2), $\hat{\theta}_k$ is the $k$th entry of the vector-valued $\hat{\theta}$, similarly $g_k$ is the $k$th entry of the vector-valued $g$, and

$$\mathrm{Var}_\theta(\hat{\theta}_k) = \mathbb{E}_\theta\left[(\hat{\theta}_k - g_k(\theta))^2\right] = \int_{\mathbb{R}^n} \left(\hat{\theta}_k(x) - \int_{\mathbb{R}^n} \hat{\theta}_k(y)\, f_\theta(y)\, dy\right)^2 f_\theta(x)\, dx \tag{3}$$

and

$$\mathbb{E}_\theta\left[\left(\frac{f_{\theta+\varepsilon}(X)}{f_\theta(X)} - 1\right)^2\right] = \int_{\mathbb{R}^n} \left(\frac{f_{\theta+\varepsilon}(x)}{f_\theta(x)} - 1\right)^2 f_\theta(x)\, dx. \tag{4}$$

If $\hat{\theta}$ is an unbiased estimator of $\theta$, then $g(\theta) = \theta$ and (2) simplifies to

$$\mathrm{Var}_\theta(\hat{\theta}_k) \geq \frac{(\varepsilon_k)^2}{\mathbb{E}_\theta\left[\left(\frac{f_{\theta+\varepsilon}(X)}{f_\theta(X)} - 1\right)^2\right]} \tag{5}$$

for $k = 1, 2, \ldots, p$ and for any vector $\varepsilon$ in the same $p$-dimensional real vector space $\mathbb{R}^p$ to which $\theta$ belongs. Requiring the estimator to be unbiased is tantamount to forbidding the use of extra, outside information such as a Bayesian prior. Unbiasedness is a reasonable yet significant restriction. If, for example, the actual values of the data are known from sources other than the observations $X$, then clearly the estimator can be better than unbiased — the estimator could simply ignore the observations $X$ and report the correct values known a-priori from another source.

A bound on the mean-square error of any unbiased estimator $\hat{\theta}$ of $\theta$ follows immediately from (5):

$$\mathbb{E}_\theta\left[\frac{1}{p}\sum_{k=1}^{p}(\hat{\theta}_k - \theta_k)^2\right] \geq \frac{\frac{1}{p}\sum_{k=1}^{p}(\varepsilon_k)^2}{\mathbb{E}_\theta\left[\left(\frac{f_{\theta+\varepsilon}(X)}{f_\theta(X)} - 1\right)^2\right]} \tag{6}$$

for any vector $\varepsilon$ in the same $p$-dimensional real vector space $\mathbb{R}^p$ to which $\theta$ belongs.

### 2.2 Dithering the features of a machine-learned model

This subsection discusses how to enhance privacy (specifically, confidentiality) of the input data used during inference with an already trained machine-learned model, by adding noise to the features that the inference calculates. The formal term for adding noise is "dithering." The present subsection specializes the HCR bounds of (2) and (5) to this setting and details algorithms for computing the bounds.

To set notation, we let the vector $\theta$ of parameters denote the input data and the vector $X$ of observations denote the resulting features, with noise added to the features. We let the vector $a_\theta$ of activations denote the features without noise added. Note that the input data need not be the entire test set, but could be only one or more of the individual examples input during inference.

Obtaining a tight HCR bound hinges on selecting a suitable vector $\varepsilon$ of perturbations, perhaps taking the maximum bound realized over several choices of $\varepsilon$. The ideal $\varepsilon$ maximizes the ratio in the HCR bound

---

**Algorithm 1:** Calculation of a perturbation $\varepsilon$ to the vector of parameters $\theta$

> **Input:** Positive integers $i$, $n$, and $p$, a vector $z$ whose $n$ entries are real numbers, a vector $\theta$ whose $p$ entries are real numbers, and functions $t$ and $u$ that apply the transpose of the Jacobian $(\partial a_\theta / \partial \theta)$ and the Jacobian itself (without transposition) to arbitrary vectors, respectively, where $a_\theta$ is the vector of features introduced in Subsection 2.2; here, $i$ is the number of repetitions of the LSQR algorithm of Paige & Saunders (1982) that the present Algorithm 1 will execute, $z$ is the starting vector for the iterations of LSQR (so $t(z)$ is the starting vector with regard to the normal equations), and $\theta$ is the unperturbed input data.

> **Output:** A vector $\varepsilon$ whose $p$ entries are real-valued and a vector $z_\varepsilon$ whose $n$ entries are real-valued; $\varepsilon$ is the perturbation to $\theta$ such that $z_\varepsilon = a_{\theta+\varepsilon} - a_\theta$.

**1** Set $z^{(0)} = z$.

**2** Calculate the vector of features $a_\theta$ corresponding to the input $\theta$.

**3 for** $j = 1, 2, \ldots, i$ **do**

**4**     Set $\tilde{z}^{(j-1)} = \|z\| \cdot z^{(j-1)} / \|z^{(j-1)}\|$, so that the Euclidean norms of $z$ and $\tilde{z}^{(j-1)}$ are equal.

**5**     Solve the least-squares problem of minimizing the Euclidean norm $\|(\partial a_\theta / \partial \theta)\, \varepsilon^{(j)} - \tilde{z}^{(j-1)}\|$, obtaining the minimizing $\varepsilon^{(j)}$ using LSQR of Paige & Saunders (1982). LSQR should invoke the functions $t$ and $u$ to perform the matrix-vector multiplications that LSQR requires. This step 5 amounts to applying the pseudoinverse of the Jacobian $(\partial a_\theta / \partial \theta)$ to $\tilde{z}^{(j-1)}$, yielding $\varepsilon^{(j)}$.

**6**     Calculate the vector of features $a_{\theta+\varepsilon^{(j)}}$ corresponding to the perturbed input $(\theta + \varepsilon^{(j)})$.

**7**     Set $z^{(j)} = a_{\theta+\varepsilon^{(j)}} - a_\theta$.

**8 end**

**9 return** $\varepsilon = \varepsilon^{(i)}$ and $z_\varepsilon = z^{(i)}$.

---

of (5). When the noise added to the features is Gaussian, with the entries of the noise being independent and identically distributed centered normal variates, the denominator in (5) becomes the expression in (12) given below. Maximizing (5) thus amounts to making the perturbation $\varepsilon$ to the input $\theta$ as large as possible while making the corresponding perturbation $z_\varepsilon$ to the features $a_\theta$ as small as possible, for $z_\varepsilon$ of (10) below; that is, the goal is to maximize the ratio of Euclidean norms $\|\varepsilon\|/\|z_\varepsilon\|$, or, equivalently, to minimize the ratio $\|z_\varepsilon\|/\|\varepsilon\|$. If the perturbation $\varepsilon$ is small, then linearization yields that $z_\varepsilon \approx (\partial a_\theta / \partial \theta)\, \varepsilon$, which is the product of the Jacobian $(\partial a_\theta / \partial \theta)$ and the perturbation $\varepsilon$ to $\theta$. Under this linear approximation, the minimum of the ratio $\|z_\varepsilon\|/\|\varepsilon\|$ is therefore equal to reciprocal of the spectral norm of the pseudoinverse of the Jacobian $(\partial a_\theta / \partial \theta)$; after all, the spectral norm of the pseudoinverse is simply the reciprocal of the least singular-value of the Jacobian itself, and the least singular-value of the Jacobian $(\partial a_\theta / \partial \theta)$ is by definition the minimum of the ratio $\|(\partial a_\theta / \partial \theta)\, \varepsilon\|/\|\varepsilon\| \approx \|z_\varepsilon\|/\|\varepsilon\|$.

Some simple iterations can approximate the spectral norm of the pseudoinverse of the Jacobian $(\partial a_\theta / \partial \theta)$ while simultaneously calculating a perturbation $\varepsilon$ for which the ratio $\|(\partial a_\theta / \partial \theta)\, \varepsilon\|/\|\varepsilon\| \approx \|z_\varepsilon\|/\|\varepsilon\|$ is nearly minimal. Indeed, iterations of LSQR of Paige & Saunders (1982) with the Jacobian $(\partial a_\theta / \partial \theta)$ applied to vectors generated during the iterations of LSQR and with the transpose $(\partial a_\theta / \partial \theta)^\top$ of the Jacobian applied to other vectors generated during the iterations can approximate the action of the pseudoinverse of the Jacobian $(\partial a_\theta / \partial \theta)$. Such iterations of LSQR produce a perturbation $\varepsilon$, from which the corresponding perturbation $z_\varepsilon$ to the features is straightforward to calculate. Then the newly calculated $z_\varepsilon$ can serve as the starting point $(\partial a_\theta / \partial \theta)^\top z_\varepsilon$ in the normal equations for further iterations of LSQR. The further iterations of LSQR yield an updated perturbation $\varepsilon$, from which the corresponding perturbation $z_\varepsilon$ to the features is straightforward to compute. Repeating this process several (say, $i = 10$) times, iteratively updating $\varepsilon$ and $z_\varepsilon$ every time, will approximately minimize the ratio $\|z_\varepsilon\|/\|\varepsilon\|$. Algorithm 1 provides pseudocode summarizing the procedure.

Note that automatic differentiation can apply to arbitrary vectors both the Jacobian and its transpose, efficiently and matrix-free (never actually having to form the full Jacobian). Furthermore, there is no need to compute $\varepsilon$ especially precisely, as any approximation whatsoever to the ideal for $\varepsilon$ yields a perfectly rigorous guarantee via the HCR bounds. Indeed, given the perturbation $\varepsilon$ to the input $\theta$, calculating the corresponding perturbation $z_\varepsilon$ to the features exactly (without any approximations) requires just one forward run of inference with the machine-learned model.

### 2.3  Additive noise

This subsection specializes the HCR bounds of the previous subsections to the case in which the dithered features are simply the features plus independent and identically distributed noise. In particular, this subsection derives an explicit expression for the right-hand side of (5). In accordance with the notation of the preceding subsection, Subsection 2.2, we denote by $a_\theta$ the features generated by inference with the already trained model using the original data $\theta$ (or just a single test example) as inputs, and we denote by $a_{\theta+\varepsilon}$ the features generated by inference using the perturbed data $(\theta + \varepsilon)$ as inputs.

Dithering yields the observed noisy vector of features

$$X = a_\theta + Z, \tag{7}$$

where $Z$ is the noise added. Denoting by $f$ the probability density function of the noise, we see that

$$f_\theta(X) = f_\theta(a_\theta + Z) = f(Z) \tag{8}$$

and

$$f_{\theta+\varepsilon}(X) = f_\theta(X - (a_{\theta+\varepsilon} - a_\theta)) = f_\theta(a_\theta + Z - (a_{\theta+\varepsilon} - a_\theta)) = f(Z - (a_{\theta+\varepsilon} - a_\theta)) = f(Z - z_\varepsilon), \tag{9}$$

where

$$z_\varepsilon = a_{\theta+\varepsilon} - a_\theta; \tag{10}$$

that is, $z_\varepsilon$ is the perturbation added to the features during determination of $\varepsilon$ (with $z_\varepsilon$ updated to correspond exactly to the $\varepsilon$ actually used, as discussed at the end of Subsection 2.2). Combining (8) and (9) yields that the denominator in (5) is

$$\mathbb{E}_\theta \left[ \left( \frac{f_{\theta+\varepsilon}(X)}{f_\theta(X)} - 1 \right)^2 \right] = \mathbb{E} \left[ \left( \frac{f(Z - z_\varepsilon)}{f(Z)} - 1 \right)^2 \right], \tag{11}$$

where $z_\varepsilon$ from (10) is viewed as a fixed constant during evaluation of the expectation.

For some distributions of $Z$ — including the multivariate normal distribution $\mathcal{N}(0, \sigma^2 \cdot \boldsymbol{I}_n)$ corresponding to a standard deviation $\sigma$ — we can evaluate (11) via analytic integration, aligning one of the axes of integration in the integral corresponding to the right-hand side of (11) with the fixed direction given by $z_\varepsilon$. The following subsection, Subsection 2.4, performs the calculation for this normal case, yielding that the denominator in (5) is

$$\mathbb{E}_\theta \left[ \left( \frac{f_{\theta+\varepsilon}(X)}{f_\theta(X)} - 1 \right)^2 \right] = \exp \left( \frac{\|z_\varepsilon\|^2}{\sigma^2} \right) - 1, \tag{12}$$

where $\|z_\varepsilon\|$ is the Euclidean norm of $z_\varepsilon$ from (10). For more complicated distributions, we can estimate the right-hand side of (11) via Monte-Carlo methods. For isotropic (that is, rotation-invariant) distributions of the added noise $Z$, such as the multivariate normal distribution $\mathcal{N}(0, \sigma^2 \cdot \boldsymbol{I}_n)$, the value of (11) depends only on the Euclidean norm of $z_\varepsilon$ and not on the entries of $z_\varepsilon$ individually. In all cases, the value of (11) is independent of the machine-learned model used.

### 2.4  Normally distributed noise

This subsection derives (12) — the case of the multivariate normal distribution in which all $n$ entries of a vector $Z$ are independent and identically distributed as $\mathcal{N}(0, \sigma^2)$, so that the probability density function (pdf) of $Z$ is

$$f(z) = \frac{1}{(2\pi\sigma^2)^{n/2}} \exp \left( -\frac{\|z\|^2}{2\sigma^2} \right), \tag{13}$$

where $\|z\|$ is the Euclidean norm of $z$. With this pdf, the right-hand side of (11) is

$$\mathbb{E} \left[ \left( \frac{f(Z - z_\varepsilon)}{f(Z)} - 1 \right)^2 \right] = \int_{\mathbb{R}^n} \left( \frac{f(z - v)}{f(z)} - 1 \right)^2 f(z) \, dz, \tag{14}$$

where $v$'s first entry $v_1 = \|z_\varepsilon\|$ is the Euclidean norm of $z_\varepsilon$ and $v$'s other entries $v_k = 0$ for $k > 1$; the invariance of (13) to rotations of the coordinate system yields (14) — the right-hand side of (14) aligns the first coordinate axis with the direction of $z_\varepsilon$. The remainder of this subsection simplifies (14) further.

Substituting (13) into the right-hand side of (14) yields

$$
\int_{\mathbb{R}^n} \left( \frac{f(z-v)}{f(z)} - 1 \right)^2 f(z)\, dz
$$

$$
= \frac{1}{(2\pi\sigma^2)^{n/2}} \int_{-\infty}^{\infty} \cdots \int_{-\infty}^{\infty} \left( \exp\left( \frac{(z_1)^2 - (z_1 - v_1)^2}{2\sigma^2} \right) - 1 \right)^2 \exp\left( -\frac{(z_1)^2 + \cdots + (z_n)^2}{2\sigma^2} \right) dz_1\, dz_2 \cdots dz_n
$$

$$
= \frac{1}{(2\pi)^{n/2}} \int_{-\infty}^{\infty} \cdots \int_{-\infty}^{\infty} \left( \exp\left( \frac{(z_1)^2 - (z_1 - c)^2}{2} \right) - 1 \right)^2 \exp\left( -\frac{(z_1)^2 + \cdots + (z_n)^2}{2} \right) dz_1\, dz_2 \cdots dz_n, \quad (15)
$$

where

$$
c = \frac{v_1}{\sigma} = \frac{\|z_\varepsilon\|}{\sigma}. \tag{16}
$$

Expanding the square yields three terms

$$
\left( \exp\left( \frac{(z_1)^2 - (z_1 - c)^2}{2} \right) - 1 \right)^2 = \exp\left( (z_1)^2 - (z_1 - c)^2 \right) - 2\exp\left( \frac{(z_1)^2 - (z_1 - c)^2}{2} \right) + 1. \tag{17}
$$

The last term in (17) corresponds in (15) to

$$
\frac{1}{(2\pi)^{n/2}} \int_{-\infty}^{\infty} \cdots \int_{-\infty}^{\infty} \exp\left( -\frac{(z_1)^2 + \cdots + (z_n)^2}{2} \right) dz_1\, dz_2 \cdots dz_n = 1. \tag{18}
$$

The penultimate term in (17) corresponds in (15) to

$$
-\frac{2}{(2\pi)^{n/2}} \int_{-\infty}^{\infty} \cdots \int_{-\infty}^{\infty} \exp\left( -\frac{(z_1 - c)^2 + (z_2)^2 + \cdots + (z_n)^2}{2} \right) dz_1\, dz_2 \cdots dz_n
$$

$$
= -\frac{2}{(2\pi)^{n/2}} \int_{-\infty}^{\infty} \cdots \int_{-\infty}^{\infty} \exp\left( -\frac{(z_1)^2 + (z_2)^2 + \cdots + (z_n)^2}{2} \right) dz_1\, dz_2 \cdots dz_n = -2. \tag{19}
$$

The first term in the right-hand side of (17) corresponds in (15) to

$$
\frac{1}{(2\pi)^{n/2}} \int_{-\infty}^{\infty} \cdots \int_{-\infty}^{\infty} \exp\left( -\frac{2(z_1 - c)^2 - (z_1)^2 + (z_2)^2 + \cdots + (z_n)^2}{2} \right) dz_1\, dz_2 \cdots dz_n
$$

$$
= \frac{1}{\sqrt{2\pi}} \int_{-\infty}^{\infty} \exp\left( -\frac{2(z_1 - c)^2 - (z_1)^2}{2} \right) dz_1 = \frac{1}{\sqrt{2\pi}} \int_{-\infty}^{\infty} \exp\left( -\frac{(z_1)^2 - 4cz_1 + 2c^2}{2} \right) dz_1. \tag{20}
$$

Further simplification yields

$$
\frac{1}{\sqrt{2\pi}} \int_{-\infty}^{\infty} \exp\left( -\frac{(z_1)^2 - 4cz_1 + 2c^2}{2} \right) dz_1 = \frac{\exp(c^2)}{\sqrt{2\pi}} \int_{-\infty}^{\infty} \exp\left( -\frac{(z_1 - 2c)^2}{2} \right) dz_1 = \exp(c^2). \tag{21}
$$

Combining all formulas in this subsection yields that the right-hand side of (11) is

$$
\mathbb{E}\left[ \left( \frac{f(Z - z_\varepsilon)}{f(Z)} - 1 \right)^2 \right] = \exp\left( \frac{\|z_\varepsilon\|^2}{\sigma^2} \right) - 1. \tag{22}
$$

## 2.5 Cramér-Rao bounds

This subsection connects the earlier subsections with the famous approach of Cramér and Rao — a connection that the original works of Hammersley (1950) and of Chapman & Robbins (1951) note as motivation for

developing their own bounds. (The remainder of this subsection will be assuming tacitly, without further elaboration, that all derivatives required for this subsection's derivations actually exist and are continuous. Unlike the HCR bounds, the Cramér-Rao bounds pertain only to scenarios in which the derivatives do exist.)

If the perturbation $\varepsilon$ is very small, then $z_\varepsilon = a_{\theta+\varepsilon} - a_\theta$ will also be very small, with $z_0 = 0$, so

$$(z_\varepsilon)_j = \sum_{k=1}^p \frac{\partial(z_\varepsilon)_j}{\partial \varepsilon_k} \varepsilon_k + o(\|\varepsilon\|) \tag{23}$$

for $j = 1, 2, \ldots, n$, while the right-hand side of (12) becomes

$$\exp\left(\frac{\|z_\varepsilon\|^2}{\sigma^2}\right) - 1 = \frac{\|z_\varepsilon\|^2}{\sigma^2} + o(\|\varepsilon\|^2) = \frac{1}{\sigma^2} \sum_{j=1}^n ((z_\varepsilon)_j)^2 + o(\|\varepsilon\|^2), \tag{24}$$

where $(z_\varepsilon)_j$ denotes the $j$th entry of the vector $z_\varepsilon$. Combining (23) and (24) yields

$$\exp\left(\frac{\|z_\varepsilon\|^2}{\sigma^2}\right) - 1 = \frac{1}{\sigma^2} \sum_{j=1}^n \left(\sum_{k=1}^p \frac{\partial(z_\varepsilon)_j}{\partial \varepsilon_k} \varepsilon_k\right)^2 + o(\|\varepsilon\|^2). \tag{25}$$

Evaluating (25) for a perturbation $\varepsilon$ in which all entries but one — say the $k$th — are zero yields

$$\exp\left(\frac{\|z_\varepsilon\|^2}{\sigma^2}\right) - 1 = \frac{(\varepsilon_k)^2}{\sigma^2} \sum_{j=1}^n \left(\frac{\partial(z_\varepsilon)_j}{\partial \varepsilon_k}\right)^2 + o(\|\varepsilon\|^2), \tag{26}$$

where $k$ is one of the positive integers $1, 2, \ldots, p$. Naturally,

$$\frac{\partial(z_\varepsilon)_j}{\partial \varepsilon_k} = \frac{1}{\partial \varepsilon_k / \partial(z_\varepsilon)_j} \tag{27}$$

for $j = 1, 2, \ldots, n$. Combining (5), (12), and (26) and taking the limit $\varepsilon \to 0$ then yields

$$\mathrm{Var}_\theta(\hat{\theta}_k) \geq \frac{\sigma^2}{\sum_{j=1}^n (\partial(z_\varepsilon)_j / \partial \varepsilon_k)^2} = \frac{\sigma^2}{\sum_{j=1}^n 1/(\partial \varepsilon_k / \partial(z_\varepsilon)_j)^2} \tag{28}$$

for $k = 1, 2, \ldots, p$, where the latter equality in (28) follows from (27). Please note that (28) is exact, not approximate — the higher-order terms vanish in the limit $\varepsilon \to 0$. Evaluating the bound (28) for all $p$ values of $k$ requires the computation of either $p$ or $n$ gradients, where $p$ is the dimension of the space of parameters and $n$ is the dimension of the space of observations. (Taking the Jacobian of $z_\varepsilon$ with respect to $\varepsilon$ requires $n$ gradients; taking the Jacobian of $\varepsilon$ with respect to $z_\varepsilon$ requires $p$ gradients.) The inequality in (28) is known as the "Cramér-Rao bound," as elucidated by Hannun et al. (2021), for example.

## 3 Results

This section applies the methods of the previous section, Section 2, to several standard data sets and neural architectures for classifying the input images.[2] All experiments of the present section pertain to the addition of independent and identically distributed Gaussian noise to features. All bounds reported in the present section are for the standard deviations corresponding to (5); of course, the standard deviation is the square root of the variance from (5). Subsection 3.1 considers MNIST, a classic data set of $28 \times 28$ pixel grayscale scans of handwritten digits, first training a simple neural net on the standard training set and then conducting inference and computing the associated HCR bounds on the test set. Subsection 3.2 does similarly for CIFAR-10, a classic data set of $32 \times 32$ pixel color images of 10 classes, namely airplanes, birds,

---

[2]Permissively licensed open-source software that can automatically reproduce all the results reported here is available at `https://github.com/facebookresearch/hcrbounds`

boats, cars, cats, deer, dogs, frogs, horses, and trucks. Subsection 3.3 considers ImageNet, a standard data set with 1000 classes, processing images from the validation set via the conventional pre-trained neural nets, "ResNet-18" and "Swin-T," from TorchVision of TorchVision maintainers & contributors (2024). Further experiments applying ResNet-18 and Swin-T to MNIST and CIFAR-10 yield results similar to those applying these same pre-trained models to ImageNet-1000 and are therefore omitted — the HCR bounds turn out to be rather ineffective and uninteresting for the larger deep nets, ResNet-18 and Swin-T.

In the coming subsections, "Affine$_{m \times n}$" refers to a layer which multiplies the input row vector from the right by an $m \times n$ matrix whose entries are learned and adds a vector which is independent of the input (that is, a "bias") that is also learned; the dimension of the input is $m$ and the dimension of the output is $n$. "ReLU" refers to a layer which preserves unchanged every non-negative entry of the input and zeros every negative entry; the dimensions of the input and of the output are the same. "Flatten" refers to a layer which reshapes the input into a single, longer vector. "Convolution2D$_{m \times n(\text{channels}); k \times \ell(\text{kernel})}$" refers to a layer which convolves each of the $m$ channels of the input with $n$ convolutional kernels, each of size $k \times \ell$ pixels whose values are learned, and adds to the result an image which is independent of the input (that is, a "bias") that is also learned. "MaximumPooling2D$_{m \times n(\text{stride}); k \times \ell(\text{kernel})}$" refers to a layer which partitions the input into $m \times n$ blocks of pixels and replaces each block with the maximum value in the block (in this paper, the stride and size of the kernel are always the same, that is, $m = k$ and $n = \ell$); the first dimension of the output is $1/m$ times the first dimension of the input, while the second dimension of the output is $1/n$ times the second dimension of the input. "Softmax" refers to a layer which calculates the softmax of the input vector (the softmax is also known as the "Gibbs distribution"); the dimensions of the input and of the output are the same.

The weights and biases in the neural networks are the learned values; the values of the inputs, features, and class-confidences are activations (that is, values at nodes) in the neural nets. All results reported are HCR bounds maximized over 25 independent and identically distributed pseudorandom realizations of $z_\varepsilon$ in (12), obtained by running the algorithm (Algorithm 1) of Subsection 2.2 with the $n$ entries of the starting vector $z$ being proportional to the normally distributed noise added to the features. The constant of proportionality is $1/\sqrt{n}$ times the size $s$ of perturbation specified in the captions to the subfigures (these sizes are $1/200$, $1/500$, and $1/1000$ for the different subfigures, as indicated in the captions). This constant of proportionality results in the right-hand side of (12) being roughly $\exp(s^2) - 1 \approx s^2$, where $s$ is the size of the perturbation ($s = 1/200$, $s = 1/500$, or $s = 1/1000$, as specified in the subfigures' captions).

## 3.1 MNIST

This subsection reports the results of numerical experiments with the standard data set, "MNIST," a data set presented by LeCun et al. (1998). MNIST contains images of handwritten digits (0, 1, 2, . . . , 9).

To calculate features for a given input, we use the activations in the last layer of the following neural network: inputs $\rightarrow$ Flatten $\rightarrow$ Affine$_{784 \times 784}$ $\rightarrow$ ReLU $\rightarrow$ Affine$_{784 \times 784}$ $\rightarrow$ ReLU $\rightarrow$ features

There are 784 entries both in the input vector for each image and in the corresponding features. The input images are $28 \times 28$ pixels, with only a single color channel (the inputs are grayscale).

Given the features, the classifier takes the argmax of the activations (values at the nodes) in the last layer of the following neural network, passing the given features as inputs to the network: features $\rightarrow$ Affine$_{784 \times 10}$ $\rightarrow$ Softmax $\rightarrow$ class-confidences

In all processing, we first normalize the pixels' potential values to range from 0 to 1, then subtract the overall mean 0.1037, and finally divide by the standard deviation 0.3081. When displaying images, we reverse all these normalizations.

For training, we use random minibatches of 32 examples each, over 6 epochs of optimization (thus sweeping 6 times through all 60,000 examples from the training set of MNIST). We minimize the empirical average cross-entropy loss using AdamW of Loshchilov & Hutter (2019), with a learning rate of 0.001.

On the test set of MNIST, the average accuracy for classification without dithering is 97.9% and with dithering is 95.1%.

In Figures 1 and 2, the size of the perturbation (either 1/200 or 1/1000) pertains to the Euclidean norm of $z_\varepsilon$ in (12). In the limit that the size is 0, the HCR bounds would become Cramér-Rao bounds (if the parameterizations of the neural networks were differentiable), as in (28). The results for the different sizes turn out to be reasonably similar.

Figure 1 histograms (over all examples in the test set) the magnitudes of the HCR lower bounds on the standard deviations of unbiased estimators for the original images' values. The estimates are for the Fourier modes in a discrete cosine transform (DCT) of type II, with the DCT normalized to be an orthogonal linear transformation (meaning real and unitary or isometric). The modes of the DCT form an orthonormal basis suitable as a system of coordinates; note that these modes are for the normalized input images, standardized such that the standard deviation of the normalized pixel values is 1 and the mean is 0. The histograms in the rightmost column of Figure 1 consider only the $8 \times 8$ lowest-frequency modes, whereas the histograms in the leftmost column consider all $28 \times 28$.

Figure 1 shows that the bounds would have been reasonably effective had the pixels of the original images not been mostly almost pure black or pure white (so that rounding away the obtained bounds denoises the estimates very effectively).

Figure 2 visualizes the HCR bounds on three examples from the test set. The visualization involves (1) adding to the modes of the DCT for the normalized original image the product of independent and identically distributed Rademacher variates (which are $-1$ with probability $1/2$ and $+1$ with probability $1/2$) times the corresponding HCR bounds, (2) inverting the DCT, and (3) reversing the per-pixel normalization back into the conventional perceptual space in which the values of pixels can range from 0 to 1 (while clipping negative values to 0 and clipping values exceeding 1 to 1). Figure 2 illustrates that the obtained bounds are significant yet ineffective (mostly since thresholding the grayscale images to purely black-and-white would denoise away much of the displayed perturbations). In Figure 2, the noisy images (a), (b), (d), (e), (g), and (h) depict the best any adversary can reconstruct the noiseless inputs (c), (f), and (i) via an unbiased estimator.

## 3.2 CIFAR-10

This subsection presents the results of numerical experiments with the standard benchmark data set, "CIFAR-10," of Krizhevsky (2009). CIFAR-10 contains images representing ten labeled classes — airplanes, birds, boats, cars, cats, deer, dogs, frogs, horses, and trucks.

To calculate features for a given input, we use the activations in the last layer of the following neural network, adapted from the net of Shahrestani (2021): inputs $\to$ Convolution2D$_{3 \times 32\text{(channels)}; 3 \times 3\text{(kernel)}}$ $\to$ ReLU $\to$ MaximumPooling2D$_{2 \times 2\text{(stride)}; 2 \times 2\text{(kernel)}}$ $\to$ Convolution2D$_{32 \times 1024\text{(channels)}; 5 \times 5\text{(kernel)}}$ $\to$ ReLU $\to$ MaximumPooling2D$_{3 \times 3\text{(stride)}; 3 \times 3\text{(kernel)}}$ $\to$ Convolution2D$_{1024 \times 3072\text{(channels)}; 3 \times 3\text{(kernel)}}$ $\to$ ReLU $\to$ Flatten $\to$ Affine$_{3072 \times 3072}$ $\to$ ReLU $\to$ features ("Flatten" simply removes dimensions that are 1 in the shape of the tensor, in this particular case.)

There are 3,072 entries both in the input vector for each image and in the corresponding features. The input images are $32 \times 32$ pixels, with three color channels (red, green, and blue).

Given the features, the classifier takes the argmax of the activations (values at the nodes) in the last layer of the following neural network, passing the given features as inputs to the network: features $\to$ Affine$_{3072 \times 10}$ $\to$ Softmax $\to$ class-confidences

In all processing, we normalize the pixels' potential values to range from 0 to 2 and then subtract 1, so that the resulting pixel values can range from $-1$ to 1. When displaying images, we reverse all these normalizations.

For training, we use random minibatches of 32 examples each, over 7 epochs of optimization (thus sweeping 7 times through all 50,000 examples from the training set of CIFAR-10). We use the AdamW optimizer of Loshchilov & Hutter (2019) with a learning rate of 0.001, minimizing the empirical average cross-entropy.

On 2,500 examples drawn at random without replacement from the test set of CIFAR-10, the average accuracy for classification without dithering is 70% and with dithering is 50%.

In Figures 3 and 4, the size of the perturbation (either $1/500$ or $1/1000$) pertains to the Euclidean norm of $z_\varepsilon$ in (12). The size $1/1000$ is close to the limit in which HCR bounds would become Cramér-Rao bounds (if the parameterizations of the neural networks were differentiable), as in (28). The results for the different sizes are quite similar.

Figure 3 histograms (over 2,500 examples from the test set) the magnitudes of the HCR lower bounds on the standard deviations of unbiased estimators for the original images' values. The estimates are for the Fourier modes in a discrete cosine transform (DCT) of type II, with the DCT normalized to be orthogonal (meaning real and unitary or isometric). The modes of the DCT form an orthonormal system of coordinates; note that these modes are for the normalized input images, standardized such that the normalized pixel values range from $-1$ to $1$. The histograms in the rightmost column of Figure 3 consider only the $8 \times 8$ lowest-frequency modes, while the histograms in the leftmost column consider all $32 \times 32$.

Figure 4 visualizes the HCR bounds on three examples from the test set. As with Figure 2, the visualization involves (1) adding to the values of the modes in the DCT for the normalized original image the product of independent and identically distributed Rademacher variates with the corresponding HCR bounds on the standard deviations, (2) inverting the DCT, and (3) reversing the per-pixel normalization back to where the values of pixels in each color channel can range from 0 to 1 (clipping negative values to 0 and clipping values that are greater than 1 to 1). In Figure 4, the noisy images (a), (b), (d), (e), (g), and (h) illustrate the best any adversary could reconstruct the noiseless inputs (c), (f), and (i) via an unbiased estimator.

Both Figure 3 and Figure 4 show that the bounds are on the precipice of guaranteeing that decent reconstructions of the original images are impossible from the dithered features.

### 3.3 ImageNet-1000

This subsection presents the results of numerical experiments with the popular data set, "ImageNet-1000," of Russakovsky et al. (2015). ImageNet-1000 contains a thousand labeled classes, each consisting of images representing a particular noun (such as a species or a dog breed).

All examples of the present subsection consider 128 examples from the validation set of ImageNet-1000, drawing the examples uniformly at random without replacement. These 128 examples are more than sufficient to find that the HCR bounds for ImageNet are ineffective, allowing the dithered features to lead to full reconstructions that are imperceptibly different from the input images. All models of the present subsection are trained on the training set of ImageNet; we downloaded the pre-trained networks from PyTorch's "model zoo" of TorchVision maintainers & contributors (2024). The input images get resized to be $91 \times 91$ pixels (with three color channels — red, green, and blue) and then upsampled to be $224 \times 224$ pixels (with the same RGB color channels) for input to the pre-trained neural nets. There are slightly fewer degrees of freedom in an image that has $91 \times 91$ pixels for each of three color channels than in the features of either of the pre-trained networks ("ResNet-18" and "Swin-T") considered here. For pre-processing, we applied to the input images the usual normalizations from the model zoo of TorchVision maintainers & contributors (2024).

In Figures 5 and 6, the size (either $1/500$ or $1/1000$) of the perturbation pertains to the Euclidean norm of $z_\varepsilon$ in (12). The size $1/1000$ is close to $0$ — close to the limit in which HCR bounds would become Cramér-Rao bounds (if the parameterizations of the neural networks were differentiable), as in (28). The results of the different sizes are similar.

The following two subsubsections refrain from displaying analogues of Figures 2 and 4, since visualizations of which reconstructions are possible (analogous to those of Figures 2 and 4) turn out to be perceptually indistinguishable from the original images.

### 3.3.1 ResNet-18

This subsubsection uses the ResNet-18 of He et al. (2016). There are 24,843 entries in the input vector for each image and 25,088 entries in the corresponding features. The average (top-1) accuracy of classification without dithering is 57% and with dithering is 54%.

Figure 5 histograms (over 128 examples from the validation set) the magnitudes of the HCR lower bounds on the standard deviations of unbiased estimators for the original images' values. The estimates are for the Fourier modes in an orthogonal discrete cosine transform (DCT) of type II. The modes of the DCT form an orthonormal system of coordinates; note that these modes are for the normalized input images, standardized such that the standard deviation of the normalized pixel values is about 1 and the mean is roughly 0. The rightmost histograms in Figure 5 consider only the $32 \times 32$ lowest-frequency DCT modes.

The bounds reported in Figure 5 are useless for all practical purposes, providing next to no guarantee of any protection against reconstruction attacks.

### 3.3.2  Swin-T

This subsubsection uses the Swin-T of Liu et al. (2021). There are 24,843 entries in the input vector for each image and 37,632 entries in the corresponding features. The average (top-1) accuracy of classification without dithering is 64% and with dithering is 54%.

Figure 6 histograms (over 128 examples) the magnitudes of the HCR lower bounds on the standard deviations of unbiased estimators for the original images' values. The estimates are for the modes in an orthogonal discrete cosine transform (DCT) of type II. The modes of the DCT constitute an orthonormal basis appropriate for a system of coordinates; note that these modes are for the normalized input images, standardized such that the standard deviation of the normalized pixel values is around 1 and the mean is approximately 0. The rightmost histograms in Figure 6 filter down to the $32 \times 32$ lowest-frequency modes.

As with Figure 5, the bounds reported in Figure 6 provide effectively no guarantee of protection against reconstructing the input images.

## 4  Conclusion

The guarantees provided by the Hammersley-Chapman-Robbins (HCR) bounds in the results presented above are sometimes on the precipice of being very useful, but are far from ideal. Theoretical understanding of why could be an interesting direction for future research. The results above consider only examples in which the neural networks are at least somewhat deep. The HCR bounds might be more useful a-priori for shallow neural-networks such as those corresponding to popular generalized linear models. However, for such models Cramér-Rao bounds are easy to calculate and simpler than the HCR analogues; none of the computational sophistication developed in the present paper is necessary to compute ideal Cramér-Rao bounds in such cases. Hannun et al. (2021) took this approach.

Thus, the HCR approach appears to be ineffectual on its own. Perhaps the best use of the HCR bounds would be to supplement other, cruder techniques for enhancing privacy. An obvious such cruder technique would be to limit the sizes of the vectors of features. In the examples considered above, the sizes of the vectors of features are never less than the corresponding numbers of pixels in the images times the numbers of color channels. In the complete absence of noise, reconstructing the whole original images from the calculated features can be possible only when the number of features is no less than the number of pixel values being reconstructed (though, even then, computational cost might limit the feasibility of full reconstruction in practice). In the presence of noise, the HCR bounds rigorously limit the quality of the reconstruction. Yet the above results indicate that the bounds are fairly ineffectual for large models. A more effective strategy than relying exclusively on the HCR bounds could be to limit the sizes of the vectors of features. After all, the numbers of degrees of freedom in the original images that any scheme whatsoever can reconstruct from the corresponding features obviously cannot ever be greater than the sizes of the vectors of features. Dithering and the HCR bounds can nicely complement the limiting of the sizes of the vectors of features.

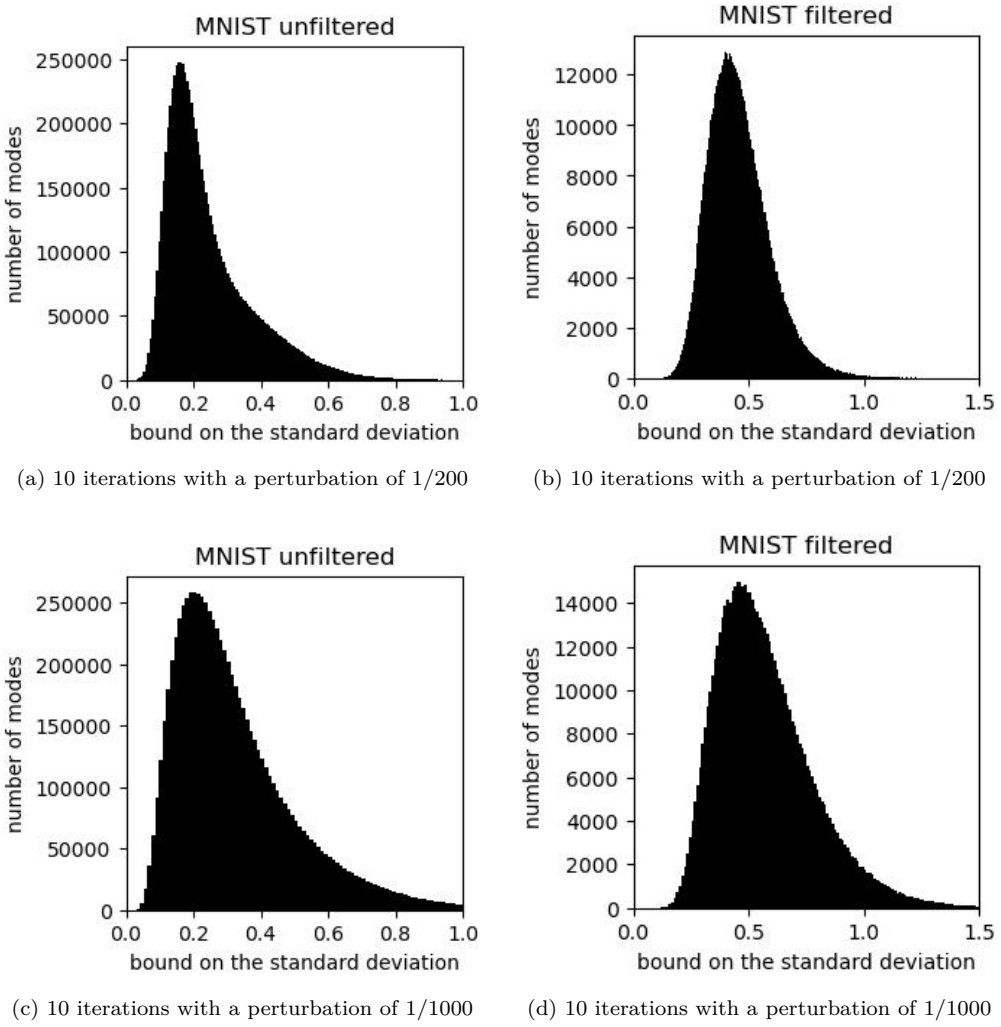

(a) 10 iterations with a perturbation of 1/200

(b) 10 iterations with a perturbation of 1/200

(c) 10 iterations with a perturbation of 1/1000

(d) 10 iterations with a perturbation of 1/1000

Figure 1: Histograms of the HCR bounds over the 10,000 examples of MNIST's test set, both unfiltered and filtered to the $8 \times 8$ lowest-frequency modes of the type-2 discrete cosine transform; the numbers of iterations are the numbers of repetitions of LSQR in Subsection 2.2, which is the input $i$ in Algorithm 1

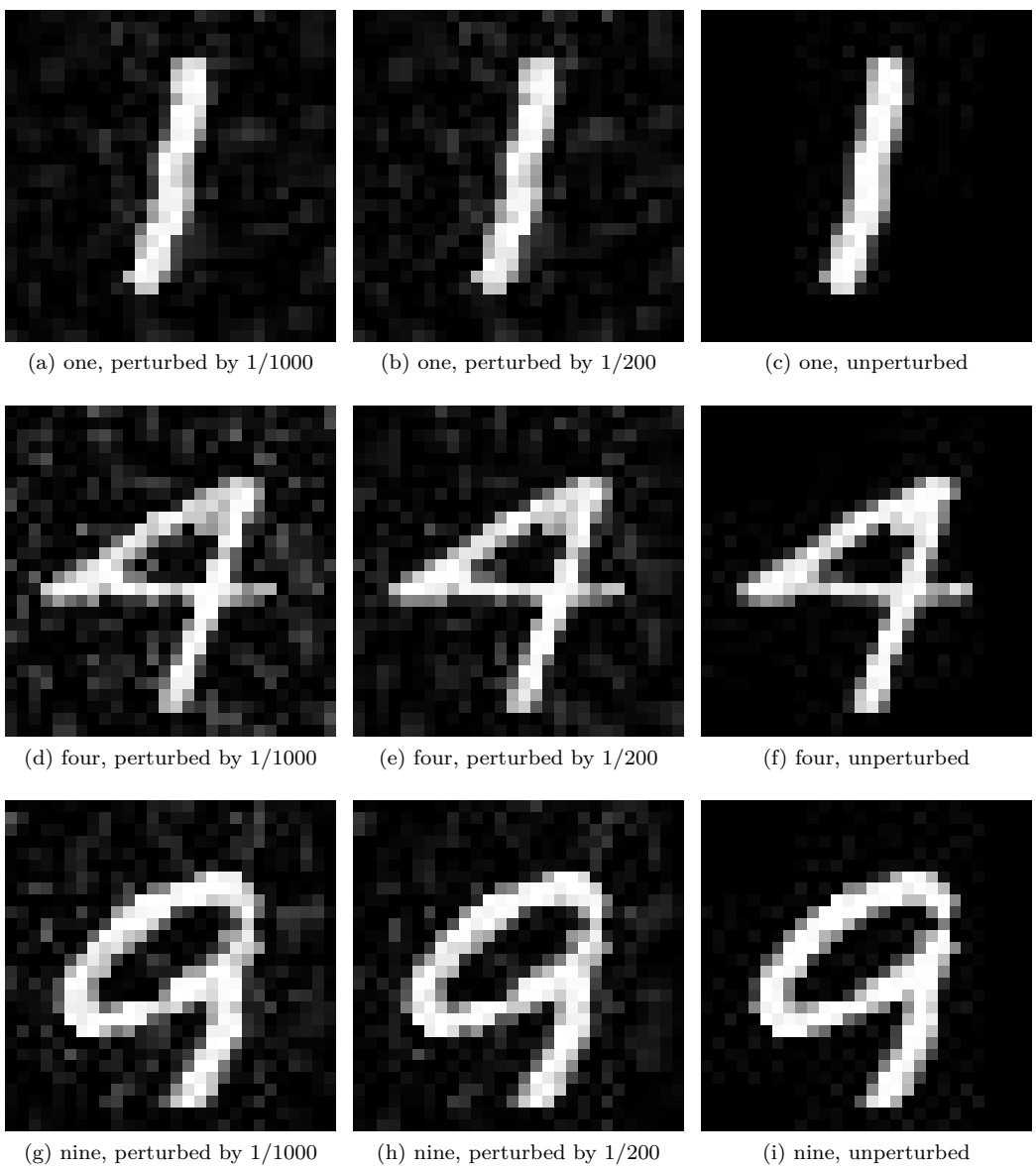

(a) one, perturbed by 1/1000     (b) one, perturbed by 1/200     (c) one, unperturbed

(d) four, perturbed by 1/1000     (e) four, perturbed by 1/200     (f) four, unperturbed

(g) nine, perturbed by 1/1000     (h) nine, perturbed by 1/200     (i) nine, unperturbed

Figure 2: Reconstructions of examples from MNIST's test set

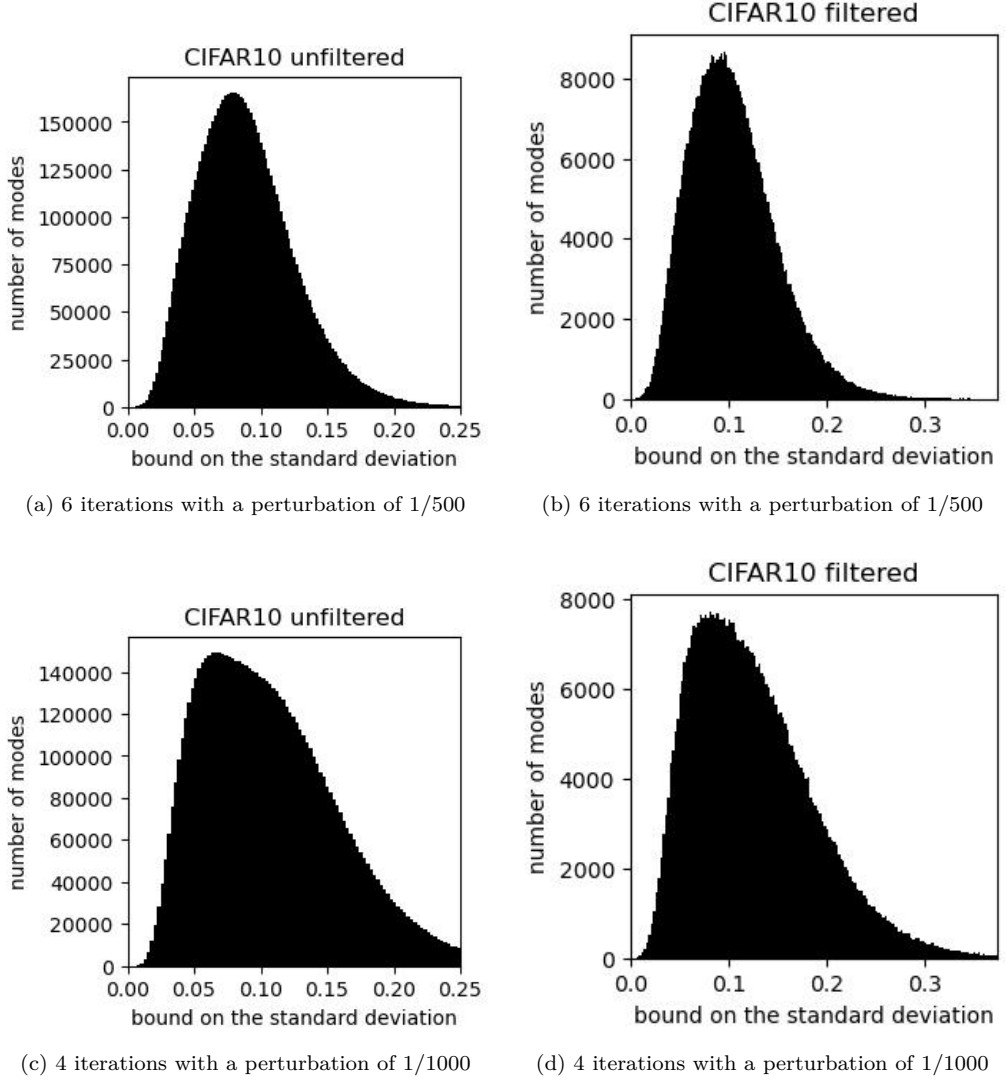

(a) 6 iterations with a perturbation of 1/500    (b) 6 iterations with a perturbation of 1/500

(c) 4 iterations with a perturbation of 1/1000    (d) 4 iterations with a perturbation of 1/1000

Figure 3: Histograms of the HCR bounds over 2,500 examples from CIFAR-10's test set, both unfiltered and filtered to the $8 \times 8$ lowest-frequency modes of the type-2 discrete cosine transform; the numbers of iterations are the numbers of repetitions of LSQR in Subsection 2.2, which is the input $i$ in Algorithm 1

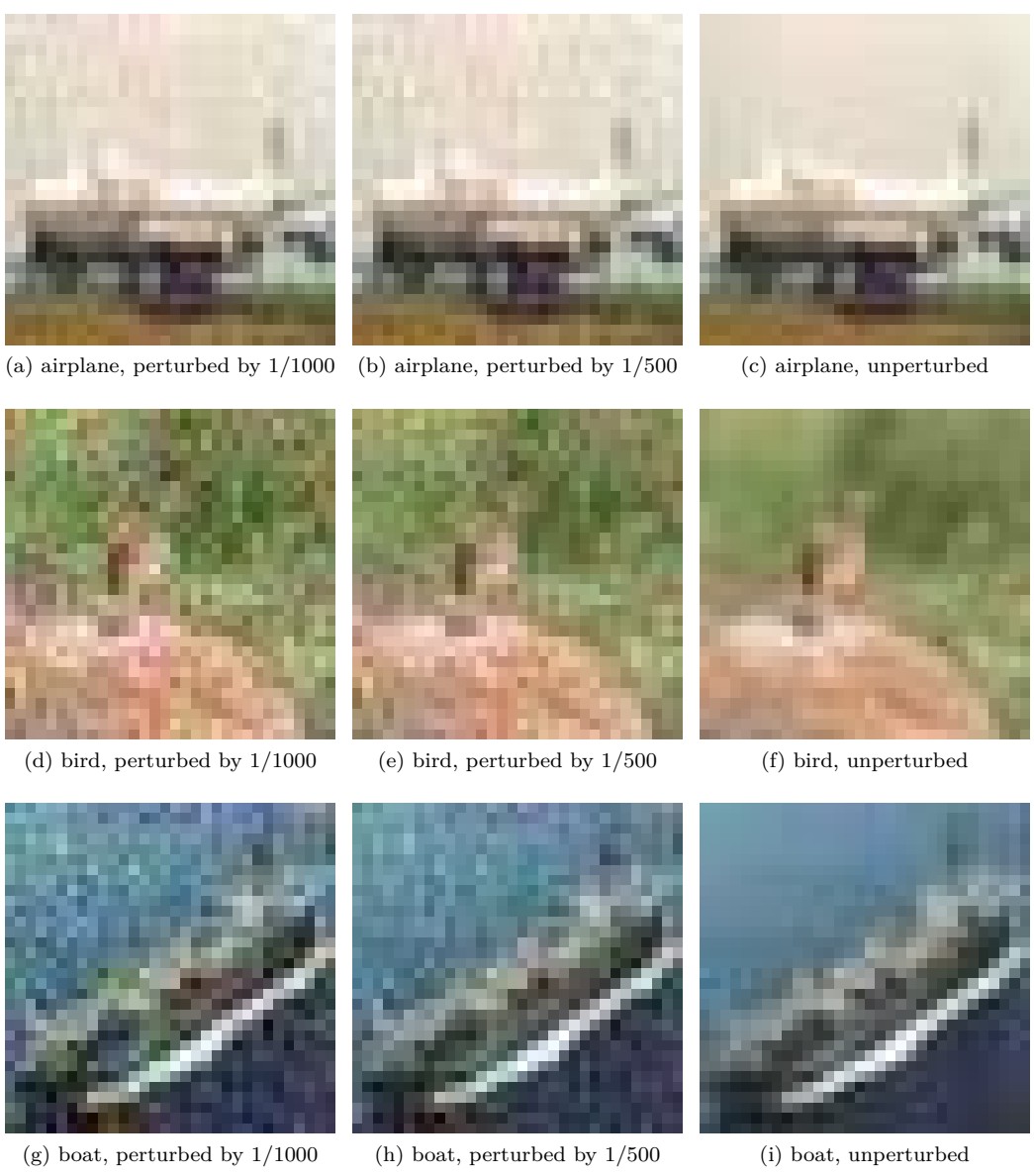

Figure 4: Reconstructions of examples from CIFAR-10's test set

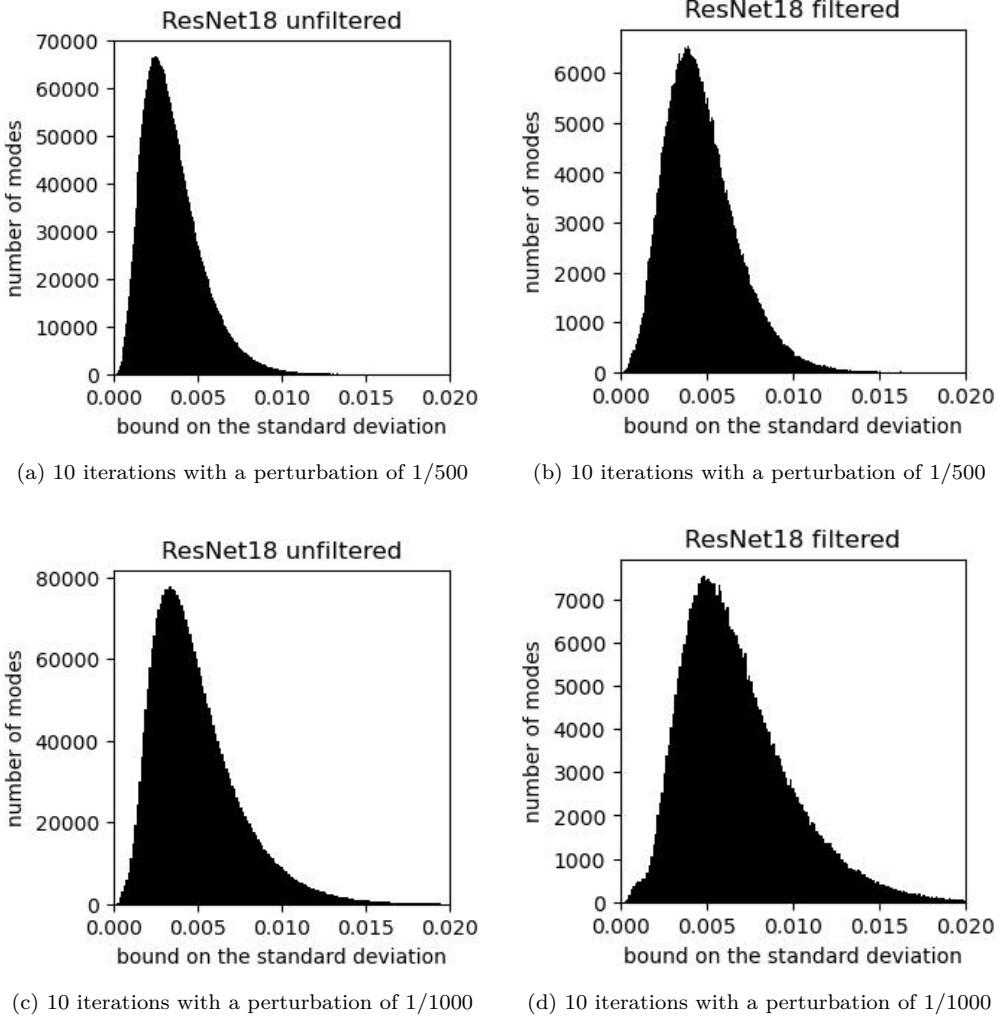

(a) 10 iterations with a perturbation of 1/500

(b) 10 iterations with a perturbation of 1/500

(c) 10 iterations with a perturbation of 1/1000

(d) 10 iterations with a perturbation of 1/1000

Figure 5: Histograms of the HCR bounds over 128 examples from ImageNet's validation set, using a ResNet-18, both unfiltered and filtered to the $32 \times 32$ lowest-frequency modes of the type-2 discrete cosine transform; the numbers of iterations are the numbers of repetitions of LSQR in Subsection 2.2, which is the input $i$ in Algorithm 1

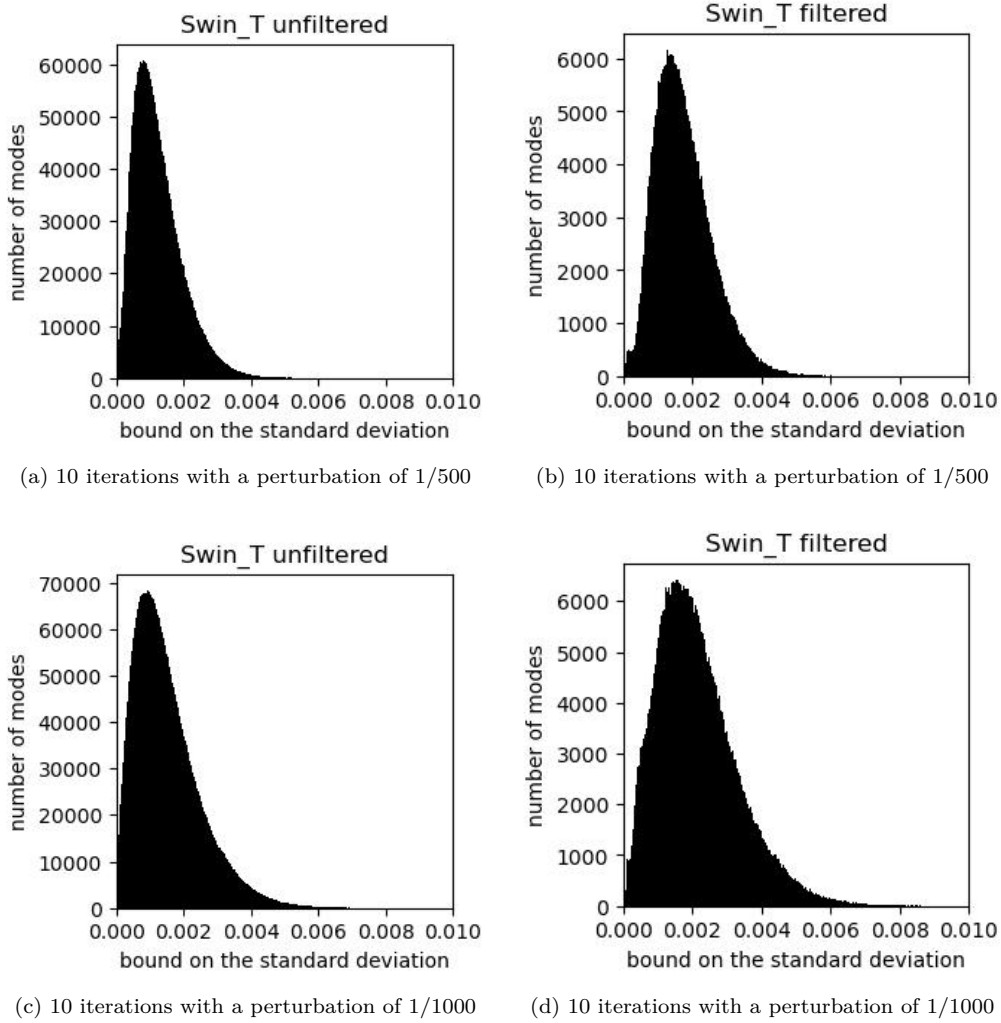

(a) 10 iterations with a perturbation of 1/500      (b) 10 iterations with a perturbation of 1/500

(c) 10 iterations with a perturbation of 1/1000      (d) 10 iterations with a perturbation of 1/1000

Figure 6: Histograms of the HCR bounds over 128 examples from ImageNet's validation set, using a Swin-T, both unfiltered and filtered to the $32 \times 32$ lowest-frequency modes of the type-2 discrete cosine transform; the numbers of iterations are the numbers of repetitions of LSQR in Subsection 2.2, which is the input $i$ in Algorithm 1

**Broader impact concerns**

Probably the greatest risk in promoting the approach of the present paper is for overstated claims of the extent to which adding noise to features protects the confidentiality of inputs to inference with machine-learned models. For example, in the absence of any accompanying visual presentation to illuminate the raw numbers, unwary users might get deceived by the numerical bounds obtained for the ResNet-18 or Swin-T when trained and run on ImageNet-1000 as in Subsection 3.3 above. Indeed, the users might not realize that the obtained bounds are largely vacuous, especially if they fail to think about what the numbers actually mean (that is, just because the bounds give trustworthy, correct, hard numbers does not mean that the numbers are useful). As shown in Subsection 3.3, the bounds do not rule out making reconstructions of the inputs to inference that are imperceptibly different from the original inputs, even when adding the greatest possible amount of noise to the features which still has negligible impact on the resulting accuracy of classification.

A technique for enhancing confidentiality (such as adding noise) should not be claimed to be fully "privacy-preserving" when in fact the resulting protection of privacy is weak or is strong only with respect to certain narrow concerns such as re-identification. Unfortunately, misleading, deceptive overselling is common in "privacy washing," as defined and discussed, for example, by Mozilla Foundation (2023). All that said, this risk from Hammersley-Chapman-Robbins bounds is likely minimal, since the bounds are so easy to interpret, directly bounding the variance of any unbiased estimator whatsoever ... and anyone can understand variance.

Still, the most popular means of preserving confidentiality may be the old standby, encryption. Any claims of protecting privacy via adding noise to features without harming the accuracy of classification should make clear that the level of protection attained is unlikely to match what cryptography could provide. Dithering features does not yield the same level of confidentiality as the familiar cryptosystems and should not be misrepresented as providing as much privacy as the ubiquitous gold-standard, encryption.

**Acknowledgements**

We would like to thank Awni Hannun, Edward Suh, the editors, and the reviewers.

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
