# OpenReview forum: "Guarantees of confidentiality via Hammersley-Chapman-Robbins bounds"
_TMLR — Accepted by TMLR_

### Review · Reviewer_9A9a · 2024-05-28

**Summary Of Contributions:**

The paper "Guarantees of Confidentiality via Hammersley-Chapman-Robbins Bounds" explores the application of Hammersley-Chapman-Robbins (HCR) bounds to quantify privacy in neural network inferences. By adding noise to the final layers of neural networks, the authors aim to prevent the reconstruction of inputs from noisy features. They demonstrate the application of HCR bounds to several standard datasets, including MNIST, CIFAR-10, and ImageNet-1000, and provide both theoretical formulations and experimental results. The study concludes that while HCR bounds can offer some privacy guarantees, they are insufficient on their own for more complex models and larger datasets.

**Audience:**

Yes

**Broader Impact Concerns:**

The Broader Impact Statement section is included in this paper, and it is sufficient.

**Claims And Evidence:**

Yes

**Requested Changes:**

Critical Changes:

1. Explore Alternative Noise Addition Techniques: Investigate and compare other noise addition methods that could complement the HCR bounds, particularly for complex models and large datasets.

2. Improve Visual Representations: Provide clearer and more impactful visualizations of the confidentiality guarantees to better illustrate the practical implications of the findings.

Strengthening Changes:

1. Detailed Comparison with Differential Privacy: Include a more in-depth comparison of the HCR bounds with differential privacy techniques to highlight relative strengths and weaknesses.

2. Expanded Discussion on Practical Limitations: Offer a more detailed discussion on the practical limitations of the HCR bounds and potential ways to mitigate these issues.

**Strengths And Weaknesses:**

Strengths:

1. Theoretical Rigor: The paper provides a detailed theoretical framework for applying HCR bounds to neural network features, showing a solid understanding of statistical bounds and their implications for privacy.

2. Practical Relevance: The study addresses a crucial aspect of machine learning—privacy during inference—offering insights into practical applications for protecting sensitive data.

3. Comparative Analysis: By comparing HCR bounds with other privacy-preserving techniques like the Cramér-Rao bound, the paper situates its contributions within the broader context of privacy research.

4. Empirical Validation: The authors conduct extensive numerical experiments on well-known datasets, providing empirical evidence to support their theoretical claims.

Weaknesses:

1. Limited Effectiveness for Complex Models: The paper admits that HCR bounds alone are insufficient for guaranteeing confidentiality in more complex models and datasets, such as ResNet-18 and Swin-T on ImageNet-1000.

2. Over-Reliance on Noise Addition: The approach heavily relies on adding noise, which may not be the most efficient or practical method for all applications. The noise levels that maintain accuracy also limit the effectiveness of privacy guarantees.

3. Lack of Generalization: The results are specific to the datasets and models used in the experiments. It is unclear how the approach would perform on other datasets or in different contexts.

4. Vague Practical Recommendations: While the paper suggests supplementing HCR bounds with other techniques, it lacks concrete guidelines or strategies for effectively combining these methods.

5. Inadequate Visual Illustrations: While the paper includes histograms and visualizations, these do not sufficiently convey the practical impact of the bounds on confidentiality.

---

> ### Comment · Editors_In_Chief · 2024-05-30
> **Request for further feedback to help us understand the requested changes**
>
> May we ask the reviewer for further feedback about some proposed revisions that could help effect the reviewer's requested changes? In the following, the reviewer's requests begin with two greater-than signs (">>"):

---

> ### Comment · Editors_In_Chief · 2024-05-30
> **1/3**
>
> >> Explore Alternative Noise Addition Techniques: Investigate and compare other noise addition methods that could complement the HCR bounds, particularly for complex models and large datasets.
>
> We apologize for the misleading formulation of the main point of the earlier submission. The purpose of our submitted paper is to study the effects of adding independent and identically distributed Gaussian noise to the features. The case of additive Gaussian noise would seem to be the most natural and is the setting in which we were able to derive reasonably elegant results. We hope to clarify in the revision that the study of the paper is limited to the case of additive Gaussian noise. For the clarification, we would propose the following changes:
>
> In the very first sentence of the abstract, modify "Protecting privacy during inference with deep neural networks is possible by adding noise..." to "Protecting privacy during inference with deep neural networks is possible by adding Gaussian noise..." with "Gaussian" specifying the special case considered.
>
> In the sentence of the abstract beginning, "Supplementing the addition of noise..." replace just "noise" with "Gaussian noise."
>
> In the first sentence of the second paragraph of the introduction, modify "This paper studies the privacy preservation arising from adding noise..." to "This paper studies the privacy preservation arising from adding independent and identically distributed Gaussian noise...."
>
> Append to the first paragraph of Section 2, "Methods," the following sentence: "The following section, Section 3, tests the bounds with several standard data sets and machine-learned models, when adding independent and identically distributed Gaussian noise to the features."
>
> Add immediately after the first sentence in the first paragraph of Section 3, "Results," the following sentence: "All experiments of the present section pertain to the addition of independent and identically distributed Gaussian noise to features."

---

> ### Comment · Editors_In_Chief · 2024-05-30
> **2/3**
>
> >> Improve Visual Representations: Provide clearer and more impactful visualizations of the confidentiality guarantees to better illustrate the practical implications of the findings. Detailed Comparison with Differential Privacy: Include a more in-depth comparison of the HCR bounds with differential privacy techniques to highlight relative strengths and weaknesses.
>
> May we ask the reviewer to clarify what is lacking in the present visualizations? We suspect that the visualizations are actually pretty much what the reviewer would like but that our earlier descriptions and explanations were lacking. To remedy our earlier lack of explanation, we would like to propose adding the following clarifying comments to the introduction of the revision (which also compares more extensively with differential privacy, as requested):
>
> The differential privacy of [Dwork-Roth] focuses on anonymization and preventing re-identification, rather than on the full confidentiality that directly prevents accurate reconstructions of input data (which is the focus of the present paper). Differential privacy provides guarantees in the form of parameters often denoted "$\epsilon$" and "$\delta$," whereas the HCR bounds directly guarantee a minimum level of noise on the best-possible reconstructions obtainable from the dithered features. Appreciating the meanings of parameters such as $\epsilon$ and $\delta$ from the usual formulations of differential privacy may require familiarity with mathematics that ordinary users are unlikely to have mastered, such as the exponential function or the concepts of relative sizes of probabilities or of the probabilities arising from classification via randomized algorithms. The HCR bounds characterize via variance the minimum level of noise that an adversarial reconstruction could possibly attain; variance (or standard deviation) is a statistic measuring noise level, variability, spread, or dispersion that is meaningful to virtually anyone.
>
> And for those who cannot understand the notion of "variance" or who prefer to see the quality of possible reconstructions displayed visually, Figures 2 and 4 below illustrate the best-possible adversarial reconstructions. Users are welcome to look at the pictures in Figures 2 and 4 -- the noisy versions are the best any adversary who lacks prior information about the input can reconstruct the noiseless inputs. (Lacking prior information refers to the restriction that the adversary use an estimator which is statistically unbiased; of course, if the adversary knew all the pixels of the input image ahead of time via external sources of information, then a biased estimator that simply ignores all data and uses the prior information would completely compromise confidentiality. The general mathematical analysis developed in Section 2 below in principle allows for adjusting the bounds to account for prior information, though working out the full theory is well beyond the scope of the present paper.) Figures 2 and 4 effectively depict error bars on the best-possible reconstructions that any adversary could ever make.
>
> Copious advantages of Cramer-Rao bounds over the canonical formulations of differential privacy have been detailed by [Hannun-Guo-van_der_Maaten] and others. Many of these advantages pertain to the HCR bounds, too. For instance, the HCR bounds adapt to the given data, whereas the standard formulation of differential privacy provides only worst-case guarantees, valid uniformly over all possible data sets (not only for the actual data of interest). The HCR bounds thus can be tighter and more powerful than the guarantees from typical differential privacy, at least with regard to preventing reconstruction of the data. Moreover, the HCR bounds can address individual pixels of images, individual modes in Fourier representations, or other parts of input data, whereas differential privacy typically operates at a coarser granularity ("membership inference").
>
> $$ $$
>
> Furthermore, we propose to add to the last paragraph of Subsection 3.1, "MNIST," and to the penultimate paragraph of Subsection 3.2, "CIFAR-10," the following sentences:
>
> In Figure 2, the noisy images (a), (b), (d), (e), (g), and (h) depict the best any adversary can reconstruct the noiseless inputs (c), (f), and (i) via an unbiased estimator.
>
> In Figure 4, the noisy images (a), (b), (d), (e), (g), and (h) illustrate the best any adversary could reconstruct the noiseless inputs (c), (f), and (i) via an unbiased estimator.

---

> ### Comment · Editors_In_Chief · 2024-05-30
> **3/3**
>
> >> Expanded Discussion on Practical Limitations: Offer a more detailed discussion on the practical limitations of the HCR bounds and potential ways to mitigate these issues.
>
> We would like to address this issue via appending to the sixth paragraph of the introduction the explanation, "From the point of view of actual practice, the idea proposed in the conclusion, Section 4, is to restrict the dimensions of the vectors of features accessible to an adversary, which directly limits the information available to the adversary. The HCR bounds are obviously more effective for practical applications in which the amount of information available to an adversary is limited. The numerical results reported in Section 3 below illustrate how the HCR bounds are more effective with smaller sizes of the vectors of features." We would also add immediately following the first sentence of the conclusion the sentence, "Theoretical understanding of why could be an interesting direction for future research." And to the first paragraph of Section 3, "Results," we would append the sentence, "Further experiments applying ResNet-18 and Swin-T to MNIST and CIFAR-10 yield results similar to those applying these same pre-trained models to ImageNet-1000 and are therefore omitted -- the HCR bounds turn out to be rather ineffective and uninteresting for the larger deep nets, ResNet-18 and Swin-T."

---

> ### Comment · Action_Editor_E3RQ · 2024-05-30
> **Can you edit the readers to add Reviewer 9A9a?**
>
> @Authors - the comments you have posted do not list Reviewer 9A9a in the readers on your comments. Can you edit the comments to change that?

---

### Review · Reviewer_4BWa · 2024-05-29

**Summary Of Contributions:**

Adding noise to the activations in the last layers of deep neural networks can protect input privacy during inference. This method's confidentiality is quantified using HCR bounds. While HCR bounds are nearly effective for small networks using MNIST and CIFAR-10 datasets, this paper proves they are insufficient for larger models like ResNet-18 and Swin-T on ImageNet-1000.

**Audience:**

Yes

**Broader Impact Concerns:**

NA.

**Claims And Evidence:**

Yes

**Requested Changes:**

See above weaknesses.

**Strengths And Weaknesses:**

Strengths:
- The paper is well-written and easy to understand.
- It is important to demonstrate both the ineffectiveness and applicable scenarios of HCR bounds.

Weaknesses:
- There are missing scenarios with HCR bound confidentiality, such as the application to small datasets like MNIST in deep neural networks and large datasets like ImageNet-1000 in shallow networks.
- It would be beneficial to include more theoretical proof of HCR bounds' ineffectiveness in deep networks, alongside the experimental analysis.

---

> ### Author Response · Authors · 2024-06-12
> **Response to the requested changes**
>
> May we ask the reviewer for further feedback about some proposed revisions that could help effect the reviewer's requested changes? In the following, the reviewer's requests begin with two greater-than signs (">>"):

---

> > ### Author Response · Authors · 2024-06-12
> > **1/2**
> >
> > >> There are missing scenarios with HCR bound confidentiality, such as the application to small datasets like MNIST in deep neural networks and large datasets like ImageNet-1000 in shallow networks.
> >
> > To report results on the scenarios raised by the reviewer, we would like to append to the first paragraph of Section 3, "Results," the sentence, "Further experiments applying ResNet-18 and Swin-T to MNIST and CIFAR-10 yield results similar to those applying these same pre-trained models to ImageNet-1000 and are therefore omitted -- the HCR bounds turn out to be rather ineffective and uninteresting for the larger deep nets, ResNet-18 and Swin-T." As for processing ImageNet-1000 with a small shallow net, the resulting accuracy is extremely poor; deep learning really is necessary in order to yield interesting results for a data set as large as ImageNet-1000.

---

> > ### Author Response · Authors · 2024-06-12
> > **2/2**
> >
> > >> It would be beneficial to include more theoretical proof of HCR bounds' ineffectiveness in deep networks, alongside the experimental analysis.
> >
> > Developing a more theoretical proof of HCR bounds’ ineffectiveness in deep networks would be wonderful. Unfortunately, we have no idea how. In fact, I have less than zero idea how -- probably any hunch I have is actually wrong and misleading. Developing such a theory looks to be a very hard problem, not something I personally could manage. For the paper itself, we propose to add immediately following the first sentence of the conclusion the sentence, "Theoretical understanding of why could be an interesting direction for future research." Perhaps someone more adept could work out a convincing theory.

---

### Review · Reviewer_Pz2W · 2024-06-09

**Summary Of Contributions:**

Ensuring privacy at inference-time with deep neural networks can be achieved by adding noise to the activations in the last layers (features/embeddings) before the final classifiers or other task-specific layers. The features become more secure when noise is introduced and reconstruction becomes difficult. The confidentiality provided by this noise can be quantified by lower bounding the variance of every possible unbiased estimator of the inputs. Convenient and computationally manageable bounds, known as Hammersley and Chapman-Robbins (HCR) bounds, are derived from classic inequalities.  The empirical evaluation show that HCR bounds are nearly effective for small neural networks with datasets like MNIST and CIFAR-10, each containing 10 classes for image classification. The HCR bounds alone appear insufficient to guarantee the confidentiality of inputs for standard deep neural networks like ResNet-18 and Swin-T, pre-trained on the ImageNet-1000 dataset, which includes 1000 classes. In all scenarios, the results discussed here consider only those levels of added noise that minimally impact the accuracy of classification from the noisy features. Thus, the noise enhances confidentiality without significantly reducing the accuracy of image classification tasks.

**Audience:**

Yes

**Broader Impact Concerns:**

There are no broader impact concerns.

**Claims And Evidence:**

Yes

**Requested Changes:**

The reviewer suggests bringing Appendix A into the main body of the paper and moving some of the qualitative results in experiments to the appendix.

**Strengths And Weaknesses:**

+ HCR method proposed here is alternative to the use of Fisher information and the Cramér-Rao bound.  CR bound is most useful when a quadratic form specified by the Fisher information matrix is a good approximation to the expected loss (the risk function) near the parameters at which the Fisher information is evaluated. HCR bound are typically tight whether or not a quadratic form specified by Fisher information is a good approximation. HCR bounds will always exist while CR bounds exist only for sufficiently smooth loss.

+ HCR method is also computationally more efficient when compared to evaluating the Fisher information for complex ML models such as deep neural networks. The approach proposed below avoids most of the difficulties and is computationally tractable.

+ For a special case whether the dithering noise is additive and some other technical conditions, an analytical simplified expression is computed and for more general case, Monte Carlo can be used to compute the denominator.

+ A detailed experimental evaluation is conducted. In itself, the effectiveness of the HCR approach is shown to be unclear.  The paper concludes that the best use of the HCR bounds  in enhancing the privacy would be to supplement other, cruder techniques, for example, by limiting the sizes of the vectors of features. The experimental results show that the sizes of the vectors of features are more than the corresponding numbers of pixels in the images times the numbers of color channels.

The reviewer appreciates the paper's attempt to describe an observation coupled with objective evaluation that would be useful to researches in the field, rather than the usual practice of proposing a method and defending it through experiments.

---

> ### Author Response · Authors · 2024-06-12
> **Response to the requested changes**
>
> May we ask the reviewer for further feedback about some proposed revisions that could help effect the reviewer's requested changes? In the following, the reviewer's requests begin with two greater-than signs (">>"):
>
> >> The reviewer suggests bringing Appendix A into the main body of the paper and moving some of the qualitative results in experiments to the appendix.
>
> Thanks so much for the very generous review! To fulfill the request, we would like to move the appendix into the main body of the paper. In the revision, what had been the appendix would become a new subsection, Subsection 2.4. This new subsection would specialize the results of Subsection 2.3 to the case of normal variates. The revision would also adjust references to the appendix to point to the new subsection instead. Specifically, the first paragraph of Section 2 would now refer to the new subsection, as would the final paragraph of Subsection 2.3. And, of course, the new subsection would no longer refer to "this appendix" but to "this subsection."
>
> One of the other reviewers requested that we highlight the visual results more. The revision will now explain those visualizations and their implications much more explicitly and hopefully clearly. As the other reviewer desired greater focus on such practical, easily interpreted visualizations, we would be somewhat hesitant to move them to an appendix. Fortunately, with the new organization in the revision, the figures would appear nearly at the end, anyways. So perhaps creating an appendix for the figures would no longer be desirable? The more involved explanation that the other reviewer requested might justify presenting the figures in the main body of the paper? We apologize for the insufficient elaboration of the visualizations in the original submission -- the graphics are actually among the most practical and informative of all results presented; our earlier failure to elucidate their meaning was unnecessarily misleading. Hopefully the revision will make clear what the visualizations really mean and their significance.

---

> > ### Comment · Reviewer_Pz2W · 2024-06-12
> > **Thank you**
> >
> > I concur with the plan. It makes a lot of sense. Thank you!

---

### Decision · Action_Editor_E3RQ · 2024-08-09

**Recommendation:** Accept as is

**Comment:**

I apologize to the authors for the delay in reaching a decision on this paper. While all three reviewers provided their reviews in a timely fashion, one reviewer failed to ever provide an official recommendation, and I finally had to make the decision based on only two reviewers.

The two reviewers who did supply final recommendations appreciate the effort that went into the revision, including a significant reorganization of the paper, and unanimously recommend acceptance.

One reviewer also recommends a reproducibility certification, and I concur.

**Audience:**

This result will be of interest to the TMLR readership.

**Claims And Evidence:**

The primary claim of this paper is a negative one: that even under relatively favorable circumstances, dithering the features produced by a deep learning image classifier are only just on the edge of preserving the privacy of the input, that is, preventing a good reconstruction of the input image. Under slightly less favorable circumstances (e.g., a larger model), dithering on its own entirely fails to prevent high-fidelity reconstruction of the input image. The claim is well supported by a careful analysis of the HCR bounds, a comparison of the HCR and Cramér-Rao bounds, and empirical results on MNIST, CIFAR-10, and Imagenet-1000 classifiers.

---

> ### Author Response · Authors · 2024-08-13
> **Thank you!**
>
> We would like to thank all the reviewers and editors! And let us be honest: this is an absolutely awesome and amazing action editor!!

---

> ### Author Response · Authors · 2024-08-19
> **What is the meaning of “reproducibility certification”?**
>
> https://jmlr.org/tmlr/reviewer-guide.html includes the following definition:
>
> “Reproducibility Certification. This is awarded to papers whose primary purpose is reproduction of other published work. Beyond simple verification, the paper must contribute significant added value through additional baselines, analysis, ablations, or insights.”
>
> Does our submission meet that criterion for the reproducibility certification? Our work is manifestly reproducible, yet might be more about developing new methods than about verifying other published work.

---

> > ### Comment · Editors_In_Chief · 2024-08-24
> > **reproducibility certification rewards the effort put into reproduction**
> >
> > dear authors,
> >
> > "reproducibility certification" was introduced at the beginning of TMLR to reward and celebrate significant effort put in by authors to reproduce any existing result (both positive and negative), and should not be understood as a sign that this paper is only about reproducing earlier findings.
> >
> > it is however understandable that the authors may not be in favour of having "reproducibility certification", in which case we can remove it from the paper.
> >
> > please let us know what you think!
> > - k

---

> > > ### Author Response · Authors · 2024-08-24
> > > **Whatever you prefer is great for us authors!**
> > >
> > > Indeed, as the editor and reviewers have observed, we provide open-source software that replicates all results in the paper automatically. If that is enough to merit designation with the reproducibility certification, then by all means please ignore the hesitation we expressed here. We just want to avoid trampling on the intent of the certification. The software replicates results which are mainly “new” rather than existing, though you could argue that use of the Cramér-Rao bound to quantify privacy preservation has been around for a while now (it won the sole Outstanding Paper Award at UAI2021, for instance), even if evaluation of the bound had been intractable at anything approaching the scale treated in our paper. So perhaps the Cramér-Rao bound in there counts as “existing” (albeit becoming tractable to compute for the first time in the paper here)? The implications of our paper for “privacy-preserving computations” are pretty negative, for sure (and there is not much we can do about that — rigorous optimal mathematical inequalities limit what is possible, whether we like it or not). The results are unfortunately unavoidably negative, in a sense similar to how the Heisenberg Uncertainty Principle is.